# BI-RADS BERT and Using Section Segmentation to Understand Radiology Reports

**DOI:** 10.3390/jimaging8050131

**Published:** 2022-05-09

**Authors:** Grey Kuling, Belinda Curpen, Anne L. Martel

**Affiliations:** 1Department of Medical BioPhysics, University of Toronto, Toronto, ON M5S 1A1, Canada; a.martel@utoronto.ca; 2Department of Medical Imaging, Sunnybrook Research Institute, Toronto, ON M4N 3M5, Canada; belinda.curpen@sunnybrook.ca

**Keywords:** BI-RADS, BERT, deep learning, natural language processing

## Abstract

Radiology reports are one of the main forms of communication between radiologists and other clinicians, and contain important information for patient care. In order to use this information for research and automated patient care programs, it is necessary to convert the raw text into structured data suitable for analysis. State-of-the-art natural language processing (NLP) domain-specific contextual word embeddings have been shown to achieve impressive accuracy for these tasks in medicine, but have yet to be utilized for section structure segmentation. In this work, we pre-trained a contextual embedding BERT model using breast radiology reports and developed a classifier that incorporated the embedding with auxiliary global textual features in order to perform section segmentation. This model achieved 98% accuracy in segregating free-text reports, sentence by sentence, into sections of information outlined in the Breast Imaging Reporting and Data System (BI-RADS) lexicon, which is a significant improvement over the classic BERT model without auxiliary information. We then evaluated whether using section segmentation improved the downstream extraction of clinically relevant information such as modality/procedure, previous cancer, menopausal status, purpose of exam, breast density, and breast MRI background parenchymal enhancement. Using the BERT model pre-trained on breast radiology reports, combined with section segmentation, resulted in an overall accuracy of 95.9% in the field extraction tasks. This is a 17% improvement, compared to an overall accuracy of 78.9% for field extraction with models using classic BERT embeddings and not using section segmentation. Our work shows the strength of using BERT in the analysis of radiology reports and the advantages of section segmentation by identifying the key features of patient factors recorded in breast radiology reports.

## 1. Introduction

The radiology report is an invaluable tool used by radiologists to communicate high-level insights and analyses of medical imaging investigations. It is common practice to organize such an analysis into specific sections, documenting the key information taken into account to determine the final impression/opinion [1]. The analysis of this report information is important in medical image data analysis for automated large-dataset labeling in machine learning computer vision. Unfortunately, computers cannot interpret and categorize raw text, and it is infeasible to manually label a large radiology corpus that can contain billions of words. Therefore, being able to automatically extract this image information in free-text radiology reports is ideal. To solve this, many researchers use natural language processing (NLP) techniques to extract radiology report information automatically. In a systematic review by Casey et al. [2], NLP techniques are used for patient or disease surveillance, identifying disease information for classification systems, language analysis optimized to facilitate clinical decision support, quality assurance, and epidemiology cohort building. In clinical breast cancer management and screening, this could include the surveillance of benign-appearing lesions over time to determine if biopsy is needed [3], or the investigation of diagnostic utilization and yield to determine hospital resource allocation [4].

A prime opportunity for NLP applications exists in the breast radiology reports of mammograms, ultrasounds, magnetic resonance imaging (MRI) exams, and biopsy reports. This information is organized into designated sections to keep reports clear and concise. The criteria and organization of this reporting system was first formalized in the 1980s by the American College of Radiologists in the Breast Imaging Reporting and Data System (BI-RADS) [5]. In a breast radiology report, many important health indicators, including menopausal status and history of cancer, are recorded together with the purpose of the exam in a section typically called clinical indication (Cl. Ind.). These details give evidence to whether the exam is for a routine screening or a diagnostic investigation of an abnormality. Imaging findings include the presence of lesions, breast density, and background parenchymal enhancement (BPE) (specifically in breast MRI). These health indicators and imaging findings can be very useful for patient care, treatment management, and research, such as large-scale epidemiology studies. For example, breast density and BPE are factors of interest in breast cancer risk prediction [6,7]. Breast density is the ratio of radiopaque tissue to radiolucent tissue in a mammogram, or the ratio of fibroglandular tissue to fat tissue in an MRI, while BPE is the level of healthy fibroglandular tissue enhancement during dynamic contrast-enhanced breast MRI. Both of these factors have been shown to have an association with the incidence of breast cancer.

Recent advancements in NLP models, notably, the bi-directional encoder representations from transformers (BERT) model developed in 2018 by Google [8], have resulted in significant performance improvements over classic linguistic rule-based techniques and word to vector algorithms for many NLP tasks. Devlin et al. showed that BERT is able to outperform all previous contextual embedding models at text sequence classification and question-and-answering response. BERT techniques were swiftly adopted by medical researchers to build their own contextual embeddings trained specifically for clinical free-text reports, such as BioBERT [9] and BioClincal BERT [10], showing the importance of a domain-specific contextual embedding.

Growing in popularity is the concept of utilizing report section organization to better improve health indicator field extraction [11,12,13]. The BI-RADS lexicon includes a logically structured flow of sections for the title of the examination, patient history, previous imaging comparisons, technique and procedure notes, findings, impressions/opinions, and an overall exam assessment category [5]. Since this practice is so well documented and followed diligently by breast radiologists, it is an ideal dataset in which to determine whether the automatic structuring of free-text radiology reports into sections will improve health indicator field extraction. We hypothesize that, using a specialized BERT embedding trained in breast radiology and fine-tuned for section segmentation and field extraction, used in sequence, will give better performance than the classic BERT embedding fine-tuned on field extraction.

With this project, we built a new contextual embedding with the BERT architecture, called BI-RADS BERT. Our data was collected from the Sunnybrook Health Sciences Centre’s medical record archives, with research ethics approval, comprised of 180,000 free-text reports in mammography, ultrasound, MRI, and image-guided biopsy procedures performed between 2005–2020. Additionally, all pathological findings in image-guided biopsy procedures were appended to the corresponding imaging reports as an addendum. We pre-trained our model using masked language modeling on a corpus of 25 million words, and then fine-tuned the model on free-text section-segmented reports to divide reports into sections. In our exploration, we found it beneficial to use the contextual embedding in conjunction with auxiliary data (BI-RADS BERTwAux) to better understand the global report context in the section segmentation task. Then, with the section of interest in a report identified, we fine-tuned further-downstream classifiers to identify imaging modality, the purpose for the exam, mention of previous cancer, menopausal status, density category, and BPE category.

## 2. Background

### 2.1. Contextual Embeddings

NLP was initially carried out using linguistic rule-based methods [14], but these were eventually succeeded by word-level vector representations. These representations saw major success with algorithms such as word2vec [15], GloVe [16], and fastText [17]. The drawback of these word representations was the lack of contextual information from word position and local grammatical cues of words in the vicinity.

This contextual information problem was ultimately solved using the ELMo contextual embedding [18]. ELMo creates a context-aware embedding for each word in a sentence by performing pre-training using masked language modeling (MLM) and next-sentence prediction (NSP) [19]. Very soon after, BERT was developed, which uses a much larger transformer architecture and pre-training corpus to fully capture intricate semantic and contextual representations [8]. These transformer contextual embeddings have shown impressive results once fine-tuned on question and answering, name entity recognition, and textual entailment identification.

Since 2018, many successors to BERT have been developed using larger corpora and architecture sizes. RoBERTa [20] was published by Facebook, demonstrating the efficient usage of BERT with an extensive parameter grid search. They found that a larger number of parameters and the usage of MLM without NSP gave superior results. Megatron-lm [21], from NVIDIA, then showed that the application of a multi-billion-parameter BERT trained across multiple graphical processing units (GPUs) gives an even greater performance, further proving that the scaling of this method to larger models results in greater generalizability.

### 2.2. Contextual Embeddings and Section Segmentation in Medical Research and Radiology

These contextual embedding algorithms have seen quick adoption to medical industry tasks. In 2019, BioBERT [9] was published, showing the application of the BERT base in medical research analysis. This model was trained on a corpus of biomedical article abstracts retrieved from PubMed. This showed that a domain-specific BERT model performed better on medical research NLP tasks, as opposed to a classic BERT base model. This was further reinforced by BioClinical BERT [10], which exhibited a performance improvement on medical domain-specific BERT training using the MIMIC-III database of intensive care unit chart notes and discharge notes [22].

In radiology reports, CheXbert [23] was trained on MIMIC-CXR [24], and showed improved performance on extracting diagnoses from radiology reports. This model had a close alignment to radiologist performance, exhibiting the benefit of utilizing a BERT model on large-scale data cohorts where expert annotations are infeasible. In breast cancer management, Liu et al. [25] assessed the performance of a BERT model trained on a clinical corpus consisting of the encounter notes, operation records, pathology notes, progress notes, and discharge notes of breast cancer patients in China. This BERT model efficiently extracted direct information on tumor size, tumor location, bio markers, regional lymph node involvement, pathological type, and patient genealogy. This work showed the application of BERT in the analysis of breast cancer treatment reports, but did not illustrate the information retrieval of patient characteristics important to epidemiology studies of breast cancer incidence and risk assessment.

A systematic review of section segmentation was published by Pomares-Quimbaya et al. in 2019 [11]. They gave a very detailed history of the task of identifying sections within electronic health records. Their review only included 39 peer-reviewed articles, suggesting this task is under-researched in the field. Popular methods outlined in the article include rule-based line identifiers, machine learning classifiers on textual feature spaces, or a hybrid method of both. BERT was developed for section segmentation by Rosenthal et al. [26], giving very impressive results on extracting information from general electronic health records.

## 3. Materials and Methods

This section will cover details on our dataset used to train BI-RADS BERT and on the BERT pre-training procedure, the BERT model architecture, and variants for improved section segmentation performance, as well as the downstream text sequence classification tasks we evaluated. A visual outline of this paper is depicted in Figure 1.

### 3.1. Data

With research ethics board approval at Sunnybrook Health Science Centre, free-text breast radiology reports from 2005–2020 were retrieved from the electronic health records archive. This data was acquired in two datasets. Further description of database statistics can be found in Table 1. All free-text breast imaging radiology reports and biopsy procedure reports were used for the development of the BI-RADS BERT embeddings.

#### 3.1.1. Breast Imaging Radiology Reports Dataset A

Inclusion criteria included all women aged 69 and younger who have had a breast MRI exam within 2005–2020; this dataset was extracted as part of a research study focused on women participating in the Ontario High-Risk Breast Screening Program [27]. In addition to the reports relating to the screening population, patients undergoing MRI for diagnostic purposes were also included. For this dataset we received 80,648 free-text reports from 7917 patients.

#### 3.1.2. Breast Imaging Radiology Report Dataset B

An additional database of radiology reports for all women who had any type of breast imaging exam between 2014–2018 was made available from a second research study. There were 23,000 reports in dataset A that were also in dataset B; therefore, redundant reports were eliminated by exam date and patient identifier. After removing all reports that were duplicates of dataset A, we were left with an additional 98,748 free-text reports from 26,390 patients.

#### 3.1.3. Pre-Training Data

For pre-training, we used 155,000 breast radiology reports from the combined datasets. The pre-training dataset ultimately contained a corpus of 25 million words. In each report, we left punctuation, acronyms, and short-form language that is typically used by radiologists, but added periods to the end of lines or paragraphs that did not end in punctuation. We felt this improved the model’s ability to understand radiology reports with minimal text pre-processing.

#### 3.1.4. Fine-Tuning Data

For fine-tuning, we used 900 breast radiology reports from the combined datasets. We did not have a separate test set in this study and decided to perform a 5-fold cross validation experiment. For fine-tuning the dataset, annotation was performed by a clinical coordinator trained in the BI-RADS lexicon criteria and advised by a breast imaging radiologist with over 20 years of experience. For each report in the fine-tuning set, each sentence was labeled into a BI-RADS section category, such as title, history/Cl. Ind., previous imaging, imaging technique/procedure, findings/procedure notes, impression/opinion, and BI-RADS final assessment category. Each report in the fine-tuning set was labeled at the report level for modality/procedure performed, purpose for the exam (whether diagnostic or screening), a mention of the patient having a previous cancer, patient menopausal status, breast density category (mammography and MRI), and BPE category (MRI only).

### 3.2. BERT Models

#### 3.2.1. BERT Contextual Embeddings

For all BERT models, we used the base model architecture with an uncased WordPiece tokenizer [8]. For BI-RADS BERT, the WordPiece tokenizer was trained from scratch, following the WordPiece algorithm [28]. With the pre-training process, we trained BI-RADS BERT from scratch using MLM only, with a sequence length limit of 32 tokens on the breast radiology report pre-training corpus. For baseline comparison in the experimental results, we used the classic BERT base model [8] and the BioClinical BERT base model [10]. Implementation was conducted in Python 3 with the transformers library developed by HuggingFace [29], and can be found at https://github.com/gkuling/BIRADS_BERT.

Pre-training was run on Compute Canada WestGrid (Dell, Vancouver, Canada) with one NVIDIA Tesla V100 SXM2 (NVIDIA, Santa Clara, United States of America) 32 GB GPU, 16 CPU cores, and 32 GB of memory. Using sequence lengths of 32 and a batch size of 256, 150,000 iterations took 26 h to train. We observed a significantly lower training time due to the size of our training set (25 million words), and lower input sequence length. This gave us the ability to train a single batch on one GPU, which lowered the processing time by not necessitating data parallelization between multiple GPUs. This is in contrast to the baseline models that are trained on input sequence lengths of 128 tokens initially, before shifting to 512 tokens for the final 10% of iteration steps. All other training parameters were kept consistent with the BERT training procedure [8].

#### 3.2.2. BERT Classifiers

Model architectures are depicted in Figure 2. For text sequence classification tasks, we used a sequence classification head attached to the embedding latent space with a multi-class output. This includes a pooling layer connected to the first token embedding of the input text sequence, which then feeds into a fully connected layer that connects to the output classification (Figure 2A). All models compared were fine-tuned on the fine-tuning dataset, and had the same multi-class sequence classification head architecture.

When including auxiliary data into the sequence classification (Figure 2B), we used a 3-layer fully connected model ending with a Tanh activation function, resulting in a vector of 128 features. This was heuristically chosen from preliminary experiments to avoid a computationally intensive hyper-parameter grid search. The auxiliary feature vector and the embedding vector were then concatenated and fed into the multi-class sequence classification head.

### 3.3. BI-RADS Specific Tasks

For all of our fine-tuning procedures, we trained for 4 epochs with a batch size of 32, optimizer Adam with weight decay, and a learning rate of 5 × 10^−5^. For this project, we performed four sets of fine-tuning tasks: section segmentation without auxiliary data, section segmentation with auxiliary data, field extraction without section segmentation, and field extraction with section segmentation. All experiments were evaluated using a 5-fold cross-validation experiment.

#### 3.3.1. Section Segmentation with and without Auxiliary Data

This model was trained to split a report document into specific information sections that are outlined in the BI-RADS lexicon. Pre-processing entailed taking the free-text input and performing sentence segmentation to then label each sentence as belonging to a specific section, such as title, patient history or Cl. Ind., prior imaging reference, technique/procedure notes findings, impression/opinion, and assessment category.

For section segmentation with auxiliary data, a sentence from the text report was identified by BI-RADS BERT by taking the BERT contextual embedding and concatenating it with the auxiliary data encoding, which was then fed into a final classifier. The auxiliary data that was used in this task was the classification of the previous sentence in the report, the number of the given sentence that it is classifying, and the total number of sentences in the report. These global textual features were intended to capture an understanding of the flow of section organization in the BI-RADS lexicon [5].

#### 3.3.2. Field Extraction without Section Segmentation

This task involves extracting specific health indicators or imaging findings from a breast radiology report. Field extraction without section segmentation was performed by feeding the whole free-text document into the BERT classifier without narrowing the text down with the section segmenter. The architecture of the BERT for sequence classification was used for each of these tasks (Figure 2A). The specific fields that were tested are described in the following bullet points:Modality/Procedure: Identification of the imaging modality or procedure description from the title, being MG, MRI, US, biopsy, or a combination of up to three of those imaging modalities/procedures;Previous Cancer: Determination of whether the attending radiologist has mentioned if the patient has a history of cancer. We included a "Suspicious" label for examples where surgery or treatments were mentioned, but the radiologist made no direct statement of whether it was for benign or malignant disease;Purpose for Exam: Determination of the purpose for the examination, being either diagnostic screening or not stated;Menopausal Status: Description of the patient’s menopausal status, being either pre- or post-menopausal, or no mention of menopausal status;Density: Description of fibroglandular tissue in the report as fatty, scattered, heterogeneously dense, ≤75% of breast volume, dense, or not stated;Background Parenchymal Enhancement: Description of background enhancement in dynamic contrast-enhanced MRI, being minimal, mild, moderate, marked, or not stated.

#### 3.3.3. Field Extraction with Section Segmentation

When performing field extraction with section segmentation, the section segmentation task is performed first, and then the field extraction is performed on the section that contains the given field. For the field extraction with section segmentation, we performed a grid search on sequence length to observe its effect on classification. We evaluated sequence lengths of 32, 128, and 512, and these results can be found in Appendix A. We found that the optimal sequence length was task-dependent, where a sequence length of 128 is better for modality and menopausal status, while 32 is better for the rest. This result seemed logical, due to the placement of this information in the report section. With the modality task, the title describes all imaging performed. Consequently, the entirety of the text needs to be evaluated. For menopausal status, the radiologists tend to mention the information at the end of the history/Cl. Ind. section, while for the other field extraction tasks, the answer to the classification question is placed near the beginning of the report section of interest.

### 3.4. Evaluation

#### 3.4.1. Performance Metrics

We decided to use two evaluation metrics in this study. First, classification accuracy (Acc.) was used to evaluate overall performance.
$$\mathrm{Acc}.=\frac{\mathrm{TP}\;+\;\mathrm{TN}}{\mathrm{TP}\;+\;\mathrm{FP}\;+\;\mathrm{TN}\;+\;\mathrm{FN}}$$
where TP, TN, FP, and FN are true positives, true negatives, false positives, and false negatives, respectively.

Second, we implemented a general F1 measure (G.F1) based on the generalized dice similarity coefficient [30].
$$G.F1=\frac{2\sum_{i=0}^{C}w_{i}\cdot TP_{i}}{\sum_{i=0}^{C}w_{i}\cdot \left(2\cdot TP_{i}+FP_{i}+FN_{i}\right)}$$
$$w_{i}=\frac{1}{P_{i}^{2}}$$
where *C* is the class label, and *P_i_* the amount of positive cases in the test set per class *i*.

We chose this metric over the classic F1 measure because it gives a more informative performance evaluation when there are large class imbalances in the test set. The weighting of *w_i_* in G.F1 forces the F1 measure to be more sensitive to inaccuracies of classification in the minority class, which is important in our dataset because our imbalance favors reports with negative findings. These class imbalances are further depicted in Figure 3.

#### 3.4.2. Statistical Analysis

Each experiment was evaluated with a 5-fold cross validation scheme. To determine the best final model, we performed statistical significance testing with 95% confidence. We used the Mann–Whitney U test to compare the medians of different section segmenters, as the distribution of accuracy and G.F1 performance was skewed to the left (medians closer to 100%) [31]. For the field extraction classifiers, we used the McNemar test (MN-test) to compare the agreement between two classifiers [32]. The McNemar test was chosen because it has been robustly proven to have an acceptable probability of Type-I errors (not detecting a difference between the two classifiers when there is a difference). After evaluating both the configurations of field extraction explored in this paper, we performed another McNemar test to assist in choosing the best technique, either using section segmentation or not. All statistical tests were performed with p-value adjustments for multiple comparisons, testing with Bonferonni correction (B.Cor.) [33]. All statistical test results can be found in Appendix B.

## 4. Results

### 4.1. Section Segmentation

Full results are displayed in Table 2. During the five-fold cross validation, the reports were stratified by modality/procedure, and then the training set at the sentence level was further stratified by section label for training–validation splits. For each contextual embedding, all models without auxiliary data performed similarly in terms of accuracy and G.F1, but multiple comparison testing showed they were significantly different from each other, suggesting that the BioClinical BERT embedding performed the best (B.Cor. U test *p* < 0.05). Incorporating auxiliary data that is applicable to the task achieves an accuracy improvement of ∼2% across all the models. We did not find statistical significance between the three section segmentation models with auxiliary data. This suggests that auxiliary data gives sufficient information to segment the reports, regardless of the contextual embedding used.

To further investigate the effectiveness of the BI-RADS contextual embeddings compared to the baselines, we performed an ablation study to see if less training data would still be useful with a specialized BERT embedding. These results are displayed in Table 3. We can see a significant improvement of the BI-RADS BERT models over the baseline embeddings when auxiliary data is included. All models in this experiment were significantly different based on the Mann–Whitney U test (B.Cor. U test *p* < 0.05). This suggests that the BI-RADS BERT model is advantageous when the section segmentation data contains fewer than 900 radiology reports.

### 4.2. Field Extraction

#### 4.2.1. Field Extraction without Section Segmentation

Results for this experiment can be found in Table 4. Statistical significance testing showed the three models were all different from each other, with 95% confidence in the field extraction tasks of BPE, modality, purpose, and previous cancer (B.Cor. MN-test *p* < 0.05). For density, BI-RADS BERT was statistically different from BioClinical and classic BERT (B.Cor. MN-test *p* < 0.05), but BioClinical and classic BERT were not significantly different from each other. In the case of menopausal status, all three models were not significantly different from each other, based on the McNemar test.

For field extraction without using section segmentation to narrow down the report before classification, our BI-RADS BERT outperformed the baseline models in all tasks. Performances in accuracy across the board were acceptably high, the lowest performance being 87.8% in density extraction. G.F1 performance was low in general, with the lowest being 13.3% for BPE extraction, suggesting that the models have a low sensitivity for the minority classes in that given task when attempting to extract the information from the whole free-text report.

#### 4.2.2. Field Extraction with Section Segmentation

Results for this experiment can be located in Table 5. This experiment entailed using the section segmenter to locate a designated section before feeding the section sentences into the field extraction classifier. Therefore, for each task, we had varying amounts of training data for each task (except for modality, because every report has a title) because not all sections appeared in all reports. This resulted in having 613 reports with history/Cl. Ind. sections for previous cancer, menopausal status, and purpose, while having 897 reports with findings sections for density and BPE. For reports missing the report section of interest, we dropped the example from training, and we gave the test subject a label of “Not Stated” during evaluation. This way, in evaluating the different models, we could still perform pair-wise performance with the McNemar test.

Overall, this experiment resulted in higher performances than field extraction without section segmentation for classic BERT, BioClinical BERT, and the BI-RADS BERT. This was found to be statistically significant over field extraction without section segmentation with all *p*-values < 0.05by using the McNemar test.

For this experiment, we see that the BI-RADS BERT outperforms the baselines in modality, previous cancer, menopausal status, purpose, and BPE (B.Cor. MN-test *p* < 0.05), but BioClinical BERT performs the best in the density category extraction (B.Cor. MN-test *p* < 0.05). The sequence length grid search results can be found in Appendix A. With statistical significance, BI-RADS BERT’s performance is different from BioClinical and classic BERT in menopausal status, modality, and previous cancer (B.Cor. MN-test *p* < 0.05). There was no statistically significant difference between models for purpose and BPE. BioClinical BERT performed the best on density, but this was only shown to have a statistical significance over BI-RADS BERT (B.Cor. MN-test *p* < 0.05), and not classic BERT.

## 5. Discussion

This report has presented the application of a BERT embedding for report section segmentation and field extraction in breast radiology reports. With different implementations and a specialized BI-RADS BERT contextual embedding pre-trained on a large corpus of breast radiology reports, we have shown that a BERT model can be effective at splitting a report’s sentences into specific sections described in the BI-RADS lexicon; then, within those report sections, it can identify pertinent patient information and findings, such as modality used/procedure performed, record of previous breast malignancy, purpose for the exam (being either diagnostic or screening), the patient’s menopausal status, breast density category, and BPE category, specifically in breast MRI.

The improved accuracy could not have been possible without structured reporting in radiology reports [1] and the BI-RADS lexicon [5]. The section structure from the American College of Radiology’s handbook for residents instructs the radiologist to write reports as a scientific report to respond to the requesting clinician’s inquiry. To identify the information in a free-text report for a given interest, centering on a section via section segmentation gives the advantage of not searching through unnecessary details for the answer. This advantage of the BI-RADS BERT model makes it more desirable than previous methods.

It is important to note that these results support the findings by Lee at al. [9] and Sentzer et al. [10], namely, that having a specialized BERT contextual embedding in your domain gives an advantage for performing NLP tasks. Here, we have shown that breast radiology imaging reports also have a distinct style and terminology which may not show up in English text corpora, web-based corpora, biological research paper corpora, or intensive care unit reporting corpora. This improvement may be explained by the process of training the embeddings from scratch and creating a specialized tokenizer that understands phrases that are common to the domain [28]. For example, we found the word "mammogram" was split up differently depending on which embedding WordPiece tokenizer was used. This example is shown in Table 6. For the classic BERT WordPiece tokenizer, "mammogram" is split into four parts, while the BioClincal BERT splits it into three parts. Our specialized BI-RADS WordPiece tokenizer gives one part for "mammogram", as a it is the most commonly used breast imaging modality and, thus, makes the embedding more efficient at identifying these important concepts as a whole, as opposed to a combination of sequential word pieces.

Furthermore, a specialized WordPiece tokenizer gives an advantage at capturing text data into shorter sequences that contain more domain-specific information. Radiologists are taught to keep reporting concise [1], leading to many smaller sentences and statements that directly correspond to the concept the radiologist is reporting. This lower sequence length, in general, seems to result in higher performances across all the tasks (as seen in Appendix A). Even when pre-training the embedding in MLM, we trained with an input sequence of 32, which still outperformed classic BERT and BioClinical BERT trained on sequence lengths of 128 and then 512 for the final 10% of the iteration steps. Therefore, by using a smaller sequence length, the embeddings are more precise and can extract information more efficiently than when using longer sequences.

The major limitations of our project are as follows. We had a limited dataset, as this was a single institutional cohort of reports that were used to build the corpus, with a majority of them being MRI reports. Further validation on external datasets is necessary to assess generalizability. However, at present, public datasets do not exist for this specialized task. By publishing our code and embeddings, we hope to make it possible for other researchers to validate this pipeline on their own private datasets. Secondly, we chose to train the BI-RADS BERT embeddings from scratch in order to build a custom BERT embedding specialized in BI-RADS vocabulary, so the BERT embeddings were not initialized from a previous BERT embedding. Previous work suggests that double pre-training on varying datasets is highly efficient [10]. Therefore, further analyses of the gains and losses from this implementation trade-off is needed. Thirdly, we could have evaluated the field extraction models to find information in identified sections that we did not use for field extraction during training. In some cases, information generally found in the history/Cl. Ind. will be in the findings or impression sections. Identifying previous cancer, menopausal status, or purpose for the exam may be possible by looking in another report section. Our fine-tuning dataset was built to not have these discrepancies, and would not be appropriate to evaluate this. Therefore, further work on this is necessary.

Domain shift is an ongoing research problem in radiology report analysis, as recording styles change through the years. For example, the BI-RADS lexicon is currently in its fifth edition (released in 2013), and it is possible that reports generated using the fourth edition, which was released in 2003, differ significantly. Our dataset spans a 15-year period, and the majority of reports were generated using the latest edition. However, it is possible that using exam date as an auxiliary data feature could improve field extraction or section segmentation.

## 6. Conclusions

This report has shown that a domain-specific BERT embedding trained on breast radiology reports gives improved performance in NLP text sequence classification tasks, in the context of breast radiology reports, over generic BERT embeddings fine-tuned to the same tasks, such as classic BERT and BioClinical BERT. We have seen that these custom embeddings are superior to general ones in extracting health indicator information pertaining to the BI-RADS lexicon. We have further shown that the inclusion of auxiliary data, such as global textual information, can significantly improve text sequence classification in section segmentation. Our objective is to build a useful tool for large-scale epidemiological studies looking to explore new factors in the incidence, treatment, and management of breast cancer.

## Figures and Tables

**Figure 1 jimaging-08-00131-f001:**
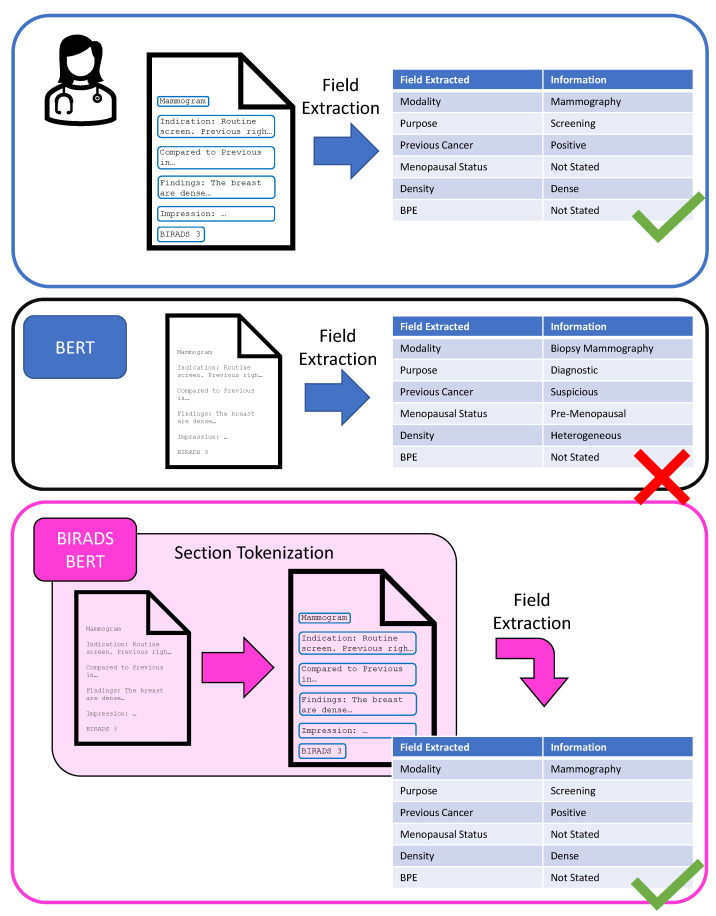
This project aims to improve health indicator field extraction tasks by using section segmentation to narrow down the free-text report length. When a radiologist reads a report, they can divide the report into sections that are useful for finding specific information. With a classic BERT framework, the report is fed into the model without narrowing the report into sections, resulting in some confusion as to where the information is located. Using a BI-RADS BERT model to segment sections before field extraction, we achieve a higher performance.

**Figure 2 jimaging-08-00131-f002:**
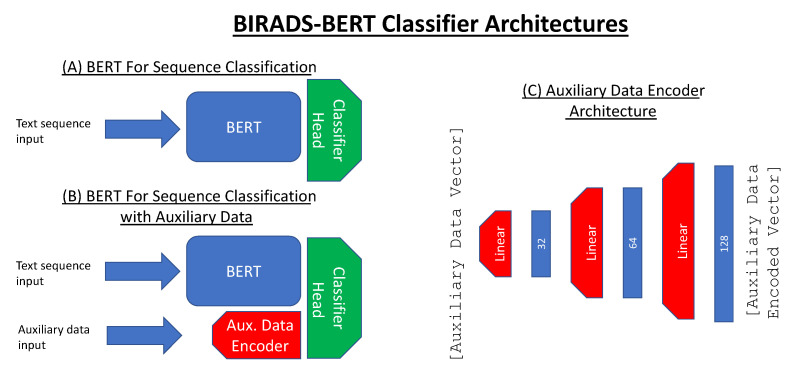
Visual representation of the model architectures used for classification. (**A**) Text sequence classifier: this model takes a contextual embedding of the input text using a BERT architecture and then feeds the embedding into a fully connected linear layer to output a classification. (**B**) Text sequence classifier with auxiliary data: this model uses a auxiliary feature encoder to build an encoded auxiliary data vector that is concatenated with the contextual embedding to use for classification. (**C**) Auxiliary data encoder architecture: this encoder architecture include 3 fully connected layers followed by a Tanh activation function.

**Figure 3 jimaging-08-00131-f003:**
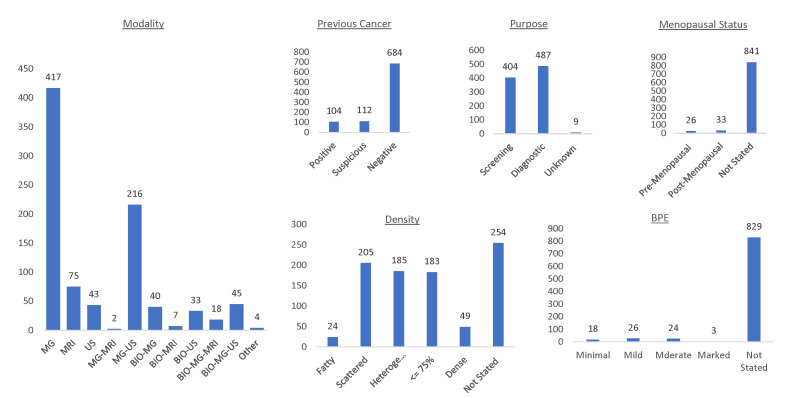
Histograms of the labels for each field that is extracted from the breast radiology reports fine-tuning dataset. We can see that each task suffers from a dominating label, which makes G.F1 better at quantifying performance over accuracy.

**Table 1 jimaging-08-00131-t001:** Dataset statistics.

Dataset Name	Number of	Number of	Avg. Exams ± St.D.	Exam
	Patients	Exams	per Patient	Date Range
Breast Imaging	7917	80,648	10.2 ± 7.0	2005–2020
Radiology Reports A				
Breast Imaging	26,390	98,748	3.7 ± 3.0	2014–2018
Radiology Reports B				

**Table 2 jimaging-08-00131-t002:** Results of 5-fold cross validation section segmentation, evaluated with average accuracy with standard deviation (Std.Dev.) and average G.F1 with standard deviation (Std.Dev.) across all 900 reports used for fine-tuning. Aux. Data = the classification of the previous sentence in the report, the number of the given sentence it is classifying, and the total number of sentences in the report.

	Base Model Fine-Tuned	Avg. Acc.	Avg. G.F1
		± Std.Dev.	± Std.Dev.
Without Aux. Data	Classic BERT	95.4±8.0%	92.1±22.6%
	BioClinical BERT	95.9±7.8%	93.2±21.3%
	BI-RADS BERT	94.1±9.5%	89.5±23.0%
With Aux. Data	Classic BERT	97.7±5.9%	94.6±20.5%
	BioClinical BERT	97.6±6.1%	94.2±21.1%
	BI-RADS BERT	**97.8 ± 5.7%**	**94.8 ± 19.8%**

**Table 3 jimaging-08-00131-t003:** Results of 5-fold cross validation section segmentation when trained using **10%** of training data, evaluated with average accuracy with standard deviation (Std.Dev.) and average G.F1 with standard deviation (Std.Dev.) across all 900 reports used for fine-tuning. Aux. Data = the classification of the previous sentence in the report, the number of the given sentence it is classifying, and the total number of sentences in the report.

	Base Model Fine-Tuned	Avg. Acc.	Avg. G.F1
		± Std.Dev.	± Std.Dev.
Without Aux. Data	Classic BERT	90.8±10.8%	82.2±31.7%
	BioClinical BERT	85.8±12.3%	60.9±41.0%
	BI-RADS BERT	92.8±10.5%	85.1±30.2%
With Aux. Data	Classic BERT	91.8±11.0%	84.8±30.9%
	BioClinical BERT	89.2±11.4%	74.7±37.0%
	BI-RADS BERT	**93.3 ± 9.9%**	**88.7 ± 25.4%**

**Table 4 jimaging-08-00131-t004:** Results of 5-fold cross validation field extraction without section segmentation, evaluated with average accuracy with standard deviation (Std.Dev.) and average G.F1 with standard deviation (Std.Dev.) across all 5 folds.

	Acc. (G. F1) of BERT Model		
	Classic	BioClinical	BI-RADS
Modality/Procedure	64.6±15.0%(23.2±6.0%)	53.8±15.0%(23.4±8.1%)	**88.7 ± 2.1% (36.7 ± 16.0%)**
Previous Cancer	75.9±0.3%(16.5±7.0%)	80.6±6.0%(33.3±24.7%)	**91.1 ± 1.1% (78.0 ± 3.0%)**
Menopausal Status	94.7±1.0%(41.9±4.7%)	94.4±1.7%(39.4±5.9%)	**95.6 ± 2.1% (55.3 ± 11.1%)**
Purpose	89.3±8.9%(6.8±2.0%)	86.6±4.0%(6.5±1.6%)	**93.9 ± 3.4% (7.1 ± 1.4%)**
Density	62.7±22.1%(26.0±8.2%)	64.4±13.3%(25.6±6.5%)	**87.8 ± 4.0% (59.0 ± 15.2%)**
BPE	86.1±10.9%(20.0±10.2%)	89.8±5.7%(19.1±6.0%)	**92.3 ± 4.3% (20.7 ± 6.5%)**

**Table 5 jimaging-08-00131-t005:** Results of 5-fold cross validation field extraction with section segmentation, evaluated with average accuracy with standard deviation (Std.Dev.) and average G.F1 with standard deviation (Std.Dev.) across all 5 folds.

	SL	Section	Training Set		Acc. (G. F1) of BERT Model	
		Used	Size (*n*)	Classic	BioClinical	BI-RADS
Modality/Procedure	128	Title	900	89.7±4.2%(27.9±7.6%)	89.7±2.8%(32.5±7.6%)	**93.8 ± 2.9% (70.6 ± 8.1%)**
Previous Cancer	32	History/Cl. Ind.	613	89.2±2.6%(83.2±3.9%)	84.2±3.4%(76.1±4.0%)	**95.1 ± 1.4% (91.8 ± 3.1%)**
Menopausal Status	128	History/Cl. Ind.	613	94.6±3.7%(61.0±25.9%)	93.1±2.2%(40.2±6.9%)	**97.4 ± 1.7% (76.1 ± 16.8%)**
Purpose	32	History/Cl. Ind.	613	94.9±3.1%(92.9±4.3%)	95.4±1.5%(93.7±2.0%)	**97.2 ± 1.1% (96.2 ± 1.5%)**
Density	32	Findings	897	94.7±3.4%(81.4±17.1%)	**95.8 ± 2.1% (89.1 ± 8.8%)**	92.7±6.0%(84.5±10.3%)
BPE	32	Findings	897	95.6±2.3%(37.5±28.6%)	96.5±2.2%(47.2±32.7%)	**97.5 ± 2.0% (84.7 ± 16.3%)**

**Table 6 jimaging-08-00131-t006:** Example of WordPiece tokenizer results for the word "mammogram".

Model	WordPiece Tokenizer Vector
Classic	[‘ma’, ‘##mm’, ‘##og’, ‘##ram’]
BioClinical	[‘ma’, ‘##mm’, ‘##ogram’]
BI-RADS	[‘mammogram’]

## Data Availability

Restrictions apply to the availability of these data. Data was obtained from Sunnybrook Health Sciences Centre and are available from the authors with the permission of Sunnybrook Health Sciences Centre.

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
