# Peer review of "BI-RADS BERT and Using Section Segmentation to Understand Radiology Reports"

_2313-433X, 2022, doi:10.3390/jimaging8050131_

Round 1

Reviewer 1 Report

The Author present an interesting method to capture data from Radiologist free text. State of the art natural language processing (NLP) domain specific contextual word embeddings have been shown to achieve impressive accuracy for these tasks in medicine, but have yet to be utilized for section structure segmentation. In this work we pre-trained a contextual embedding Bidirectional Encoder Representations from Transformers (BERT, please explain the acronyms) model using breast radiology reports and developed a classifier that incorporated the embedding with auxiliary global textual features in order to perform section segmentation. This model achieved a 98% accuracy at segregating free text reports sentence by sentence into sections of information outlined in the Breast Imaging Reporting and Data System (BI-RADS) lexicon, with a significant improvement over the Classic BERT model and was tested in a big datataset.

Despite it is not my field of investigation, I have no comments to the manuscript. I suggest only to discuss the potential of structured reporting in Radiology, that could facilitate this task.

Author Response

Thank you so much for your insightful and kind suggestion to improve this article. As per your suggestions, I have made the following edits:

I suggest only to discuss the potential of structured reporting in Radiology, that could facilitate this task.

Added in (Line 370-377) to the Discussion: The improved accuracy can be explained by the utilization of structured reporting in radiology reports, and the BI-RADS lexicon. The section structure from the American College of Radiology handbook for residents instructs the radiologist to write reports as a scientific report to respond to the requesting clinician's inquiry. To identify the information in a free text report for a given interest, centering on a section through section segmentation gives an advantage of not searching for the answer through unnecessary information. This is the advantage of the BI-RADS BERT model over previous methods.

Reviewer 2 Report

The reviewed paper pre-trained a new contextual embedding with the BERT architecture called BI-RADS BERT. The results showed the strength of BERT in radiology report analysis, such as section segmentation and field extraction. The authors also proposed to extract the key features/fields of exam reports after section segmentation to improve the performance of BI-RADS BEAT further.

I have a few comments that may improve the quality of the paper:

  • Almost haft of the data used in this study are related to the MRI exams. There should be some common words that are shared by all the MRI related reports. Do you think it will be beneficial to train different models for different types of exams?
  • Is there a separate test set in this study? If not, please clarify this. In addition, it looks like the accuracy and G.F1 score reported in table 2-5 are the results of 5-fold cross validation on the 900 training reports for fine-tuning. Please also specify it.
  • In section 3.3.3, the authors performed a grid search on sequence length to observe its effect on classification. I suggest to discuss the results of Appendix A either here or in the result section. It is necessary to clarify the optimal sequence length used in this study.
  • In section 3.4.1, what’s the meaning of P_i in the definition of G.F1 ?
  • In section 4.2.2, the performance of field extraction will be affected by the result of section segmentation task. Could you add more discussion about this? For task “Previous Cancer”, “Menopausal Status” and “Purpose”, only 613 (~ two thirds of the whole dataset) reports were included. It is hard to draw the conclusion that field extraction with section segmentation performs better than without section segmentation, since the size of dataset is much smaller. A following question is that how to extract the “Previous Cancer”, “Menopausal Status” and “Purpose” information from the reports which don’t have the section “History/Cl. Ind.” Is it possible to extract these information based on other sections, or have to use the tranined field extraction model without section segmentation ? 
  • There is a typo (“Appendix A” instead of “1”) in section 4.2.2, line 330.
  • In table 2 and 3, “Base Model Fine Tuned” should be placed in the second column.
  • The “Task” column in table 4 and 5 is not consistent. Please correct it.
  • Similar to table 2 and 3, I suggest to add the standard deviation of the accuracy and G.F1 in table 4 and 5.

Author Response

Thank you so much for your insightful and kind suggestion to improve this article. As per your suggestions, I have made the following edits:

Almost haft of the data used in this study are related to the MRI exams. There should be some common words that are shared by all the MRI related reports. Do you think it will be beneficial to train different models for different types of exams?
Response: I think that this is definitely something to explore if a researcher had access to a larger dataset of radiology reports for other imaging sites and not just breast. Where we only had breast imaging I do not believe it would be advantageous to make separate modals by imaging modalities since a lot of common language is used across imaging modalities to describe the findings of the breast. We also had multiple breast exam reports that combine imaging modality exams into one report. For example, if the patient recieved a mammogram, here was a suspicious lesion identified, and then they performed an ultrasound guided biopsy, the radiologist would combine the mammogram, ultrasound and biopsy report into one free text report. Thereofre, if you had all radiology reports for multiple sites of the body it may be more advantageous to have separate modality models since the common language across modalities would be more advantageous. Although, I think a further investigation of which choice is better would be very interesting, to have specialized models in imaging site, or imaging modality, or a combination of both. 

Is there a separate test set in this study? If not, please clarify this. In addition, it looks like the accuracy and G.F1 score reported in table 2-5 are the results of 5-fold cross validation on the 900 training reports for fine-tuning. Please also specify it.

Response: Added in (Line 166-183) in the materials and methods: Pre Training Data: For pre-training, we used 155 thousand breast radiology reports from the combined datasets. The pre-training dataset ultimately contained a corpus of 25 million words. In each report, we left punctuation, acronyms, and short form language that is typically used by radiologists, but added periods to the end of lines or paragraphs that did not end in punctuation. We felt this improved the model's ability to understand radiology reports with minimal text pre-processing. 

Fine Tuning Data: For fine-tuning, we used 900 breast radiology reports from the combined datasets. We did not have a separate test set in this study and decided to perform a 5 fold cross validation experiment. For the fine-tuning dataset annotation was performed by a clinical coordinator trained in the BI-RADS lexicon criteria and advised by a breast imaging radiologist with over 20 years experience. For each report in the fine-tuning set, each sentence was labeled into a BI-RADS section category, such as title, history/clinical indication, previous imaging, imaging technique/procedure, findings/procedure notes, impression/opinion, and BI-RADS final assessment category.  Each report in the fine-tuning set was labeled at the report level for modality/procedure performed, purpose for the exam (whether diagnostic or screening), a mention of the patient having a previous cancer, patient menopausal status, breast density category (mammography and MRI), and BPE category (MRI only).

In section 3.3.3, the authors performed a grid search on sequence length to observe its effect on classification. I suggest to discuss the results of Appendix A either here or in the result section. It is necessary to clarify the optimal sequence length used in this study.
Response: Added in (Line 262-271) Materials and Methods: We evaluated sequence lengths of 32, 128 and 512 and these results can be found in Appendix A. We found that the optimal sequence length was task dependent, where a sequence length of 128 for modality and menopausal status, while 32 is better for the rest. This result seemed logical due to the placement of this information in the report section. With the modality task, the title describes all imaging performed. Consequently, the entirety of the text needs to be evaluated. For menopausal status, the radiologists tend to mention the information at the end of the history/clinical indication section. While the other fields extraction tasks, the answer to the classification question is placed near the beginning of the report section of interest. 

In section 3.4.1, what’s the meaning of P_i in the definition of G.F1 ?

Response: Added in (Line 280) in MAterials and Methods: where C is the class label, and P_i the mount of positive cases in the test set per class i. 

In section 4.2.2, the performance of field extraction will be affected by the result of section segmentation task. Could you add more discussion about this? For task “Previous Cancer”, “Menopausal Status” and “Purpose”, only 613 (~ two thirds of the whole dataset) reports were included. It is hard to draw the conclusion that field extraction with section segmentation performs better than without section segmentation, since the size of dataset is much smaller. A following question is that how to extract the “Previous Cancer”, “Menopausal Status” and “Purpose” information from the reports which don’t have the section “History/Cl. Ind.” Is it possible to extract these information based on other sections, or have to use the tranined field extraction model without section segmentation ? 

Response: I have addressed this in two places. First a clarification in 4.2.2. Added in (Line 343-346)in the Results: For reports missing the report section of interest, we dropped the example from training, and we gave the test subject a label of "Not Stated" during evaluation. This way, in evaluating the different models, we could still perform pair wise performance with the McNemar Test. 

And later on (Line 412-418) in the Discussion: Thirdly, we could have evaluated the field extraction models to find information in identified sections that we did not use for field extraction during training. In some cases, information generally found in the history/clinical indication will be in the findings or impression sections. Identifying previous cancer, menopausal status, or purpose for the exam may be possible by looking in another report section. Our fine-tuning dataset was built not to have these discrepancies and would not be appropriate to evaluate this. Therefore, further work on this is necessary. 

The next four comments were also implemented:

There is a typo (“Appendix A” instead of “1”) in section 4.2.2, line 330.

In table 2 and 3, “Base Model Fine Tuned” should be placed in the second column.

The “Task” column in table 4 and 5 is not consistent. Please correct it.

Similar to table 2 and 3, I suggest to add the standard deviation of the accuracy and G.F1 in table 4 and 5.

Reviewer 3 Report

Thi intersting article sets out to pre-train a contextual embedding BERT model using breast radiology reports to perform section segmentation.

My main comment is that the hypothesis is not well described and should be stated plainly at the end of the hypothesis. In the current form the authors already included parts of the results section at the end of the introduction section. Actually, the authors do a better job in describing the purpose of this work in figure 1.

Additionally, I would recommend to better describe the clinical use case and implications of this work.

Please state the approval of your local ethics committee with registration number. 

Please mak it more clear how you split the data set. 

Author Response

Thank you so much for your insightful and kind suggestion to improve this article. As per your suggestions, I have made the following edits:

My main comment is that the hypothesis is not well described and should be stated plainly at the end of the hypothesis. In the current form the authors already included parts of the results section at the end of the introduction section. Actually, the authors do a better job in describing the purpose of this work in figure 1.

Response: Added in (Line 74-77) in the introduction:  We hypothesize that using a specialized BERT embedding trained in breast radiology fine tuned for section segmentation and field extraction used in sequence will give better performance than the Classic BERT embedding fine tuned on field extraction. 

Additionally, I would recommend to better describe the clinical use case and implications of this work.

Response: Added in (Line 36-39):  In clinical breast cancer management and screening, this could include surveillance of benign appearing lesions over time to determine if biopsy is needed, or the investigation of diagnostic utilisation and yield to determine hospital resource allocation. 

Please state the approval of your local ethics committee with registration number. 

Response: Added in (Line 449-452) in the Institutional Review Board Statement: The study was conducted in accordance with the Declaration of Helsinki, and approved by the Institutional Review Ethics Board of Sunnybrook Health Sciences Centre (registration number 2531 and September 23, 2019).

Please mak it more clear how you split the data set. 

Response: Added in (Line 166-183) in the materials and methods: Pre Training Data: For pre-training, we used 155 thousand breast radiology reports from the combined datasets. The pre-training dataset ultimately contained a corpus of 25 million words. In each report, we left punctuation, acronyms, and short form language that is typically used by radiologists, but added periods to the end of lines or paragraphs that did not end in punctuation. We felt this improved the model's ability to understand radiology reports with minimal text pre-processing. 

Fine Tuning Data: For fine-tuning, we used 900 breast radiology reports from the combined datasets. We did not have a separate test set in this study and decided to perform a 5 fold cross validation experiment. For the fine-tuning dataset annotation was performed by a clinical coordinator trained in the BI-RADS lexicon criteria and advised by a breast imaging radiologist with over 20 years experience. For each report in the fine-tuning set, each sentence was labeled into a BI-RADS section category, such as title, history/clinical indication, previous imaging, imaging technique/procedure, findings/procedure notes, impression/opinion, and BI-RADS final assessment category.  Each report in the fine-tuning set was labeled at the report level for modality/procedure performed, purpose for the exam (whether diagnostic or screening), a mention of the patient having a previous cancer, patient menopausal status, breast density category (mammography and MRI), and BPE category (MRI only).

Round 2

Reviewer 2 Report

Thanks for addressing my comments. I have no further question.